# Shelf Life and Functional Quality of Almond Bagasse Powders as Influenced by Dehydration and Storing Conditions

**DOI:** 10.3390/foods13050744

**Published:** 2024-02-28

**Authors:** Stevens Duarte, Ester Betoret, Noelia Betoret

**Affiliations:** Instituto Universitario de Ingeniería de Alimentos-FoodUPV, Universitat Politècnica de València, 46022 Valencia, Spain; steduase@doctor.upv.es (S.D.); mesbeval@upvnet.upv.es (E.B.)

**Keywords:** almond bagasse, air drying, freeze drying, valorization, peroxide index, plant-based byproducts

## Abstract

Almond bagasse resulting after the production of almond-based drinks represents a promising by-product with potential for use as a functional ingredient. To facilitate its utilization, the stability of this material can be achieved through dehydration processes such as hot air drying or freeze-drying. Nevertheless, owing to its high fat content, almond bagasse is prone to lipid oxidation, which could result in undesirable quality. Therefore, the objective of this work was to assess the impact of dehydration (by hot air drying at 60 and 70 °C and by freeze-drying) and storage (at room temperature and in accelerated conditions) on the functional quality and stability of almond bagasse powder. Throughout the dehydration process, it was observed that antioxidant compounds were preserved without significant differences among dehydration treatments. These compounds increased over the storage period, especially in the samples treated with hot air. Regarding antiradical capacity, the hot-air-dried samples showed higher values than the freeze-dried ones, although in all cases, it increased during storage. For total phenols in samples air-dried at 70 °C, increases of more than 50% were observed. The acidity and peroxide index were increased in the extended storage period, although they did not reach critical values. Samples stored for 180 days showed peroxide values ranging from 10 to 12.8 meq O_2_/kg dry matter for samples stored at room temperature and from 14.7 to 23 meq O_2_/kg dry matter for samples subjected to accelerated storage.

## 1. Introduction

Almond (*Prunus dulcis*) belongs to the subfamily Prunoideae, family Rosaceae (Ahmad, 2010). The kernel is a highly nutritious plant-based food, and due to the numerous health benefits associated with its regular consumption, it is gaining popularity as a healthy food [1]. Almonds are referred to as “the king of nuts” because they are amongst the most widely consumed nuts globally [2,3]. They are commercially cultivated in various countries, such as the United States, Spain, Australia, Morocco, Iran, Turkey, and Chile. Globally, their production has increased from 1.03 million metric tons in 2014–2015 to 1.48 million metric tons in 2019–2020 [4]. Specifically, almond plantations are extensively spread across the Mediterranean region, encompassing a total area of 2,162,263 hectares. Spain holds the largest area dedicated to almond cultivation, reaching 744,470 hectares [5].

Numerous studies have been conducted on the composition of almond kernels, revealing their high content of nutrients such as fatty acids, lipids, amino acids, proteins, carbohydrates, dietary fiber, and bioactive compounds such as phytosterols, polyphenols, minerals, and vitamins [6,7]. The total fat content ranges from 32% to 66% and it is mostly monounsaturated fat, providing them with remarkable oxidative stability [8]. Oleic acid predominates as the primary fatty acid, constituting 50% to 70% of the total fatty acids. Linoleic, palmitic, and stearic acids are present at levels from 10% to 26%, 5% to 9%, and 1.5% to 4%, respectively. The concentrations of linolenic and myristic acids are exceedingly low, less than 0.1% [9]. Almonds contain approximately 10–29% protein. While they are considered relatively high in protein compared to other nuts, they do not provide a complete range of essential amino acids, thus excluding them as a source of high-biological-value protein [10]. Additionally, almonds contain about 14 to 26.6% dietary fiber, primarily located in the walls of the almond kernel. The dietary fiber found in almonds has been reported to have prebiotic effects on the gut microbiota [11]. Almonds offer a variety of bioactive compounds, with phenolic compounds being the most notable. Approximately 130 phenolic compounds have been identified in almonds to date [12]. The key phenolic compounds include chlorogenic acid (9.5 mg/100 g), catechin (11.04 mg/100 g), epicatechin (12.47 mg/100 g), and isorhamnetin-3-O-rutinoside (48.5 mg/100 g) [13]. Taking into account its composition, the almond is a nut of interest both for fresh consumption and for the production of derived products.

Moreover, in recent years, plant-based beverages have gained immense popularity, either due to the increase in lactose intolerance or for ideological reasons. Among them, almond-based vegetable drinks stand out [14]. The production process is straightforward, involving a solid–liquid extraction using water as a solvent. The resulting solid residue, known as pulp or bagasse, is frequently discarded or repurposed as animal feed [15]. Nonetheless, this residue contains a significant amount of nutritional compounds such as lipids, proteins, and bioactive components.

The intriguing composition of the bagasse resulting from the production of almond-based vegetable drinks makes it a potential suitable substrate for the production of functional ingredients beneficial to the food industry [16]. Stabilizing bagasse through various dehydration methods, such as hot air drying or freeze-drying, is a valuable strategy to improve its durability and preserve its bioactive components [17]. Additionally, the combination of the drying process with a grinding operation would result in functional powdered ingredients of interest to the food industry [16]. The production of powdered ingredients with functional properties useful for the food industry is an idea that has been extensively researched in recent years. There are published studies on the use of waste or by-products from fruits, vegetables, legumes, cereals, and nuts. In the case of almonds, studies focus on the use of hulls or shells, but there are no studies on the use of bagasse. No studies have been carried out on the feasibility of obtaining powdered products from almond bagasse or on its stability during storage.

Foods or ingredients with low moisture content, high porosity, and high levels of fat and other bioactive compounds, such as almond bagasse powder, are susceptible to changes in properties when exposed to conditions of high relative humidity, high temperature, and/or the presence of light and oxygen. They may suffer from wetting and caking, oxidation reactions of fats and bioactive compounds, increased acidity, and changes in color and texture. While relative humidity, temperature, light, and oxygen are exogenous factors that promote oxidation processes, there are also endogenous factors, such as oxidative enzymes found in these natural foods, such as lipoxigenase; their activation usually occurs when plant tissue is disrupted [13]. Thus, the most common deteriorative reactions include oxidation with increasing acidity values and peroxide index. Dehydration methods and storage conditions are critical to quality degradation. Hot air drying is a process in which the food is placed in contact with a stream of hot and dry air, which facilitates oxidation reactions due to the presence of oxygen. However, it is a widely used process in the food industry, is accessible, not overly expensive, and can be used at moderately high temperatures, such as 60 and 70 °C, which minimizes damage to macromolecules and bioactive compounds. Freeze-drying is carried out under vacuum conditions, which minimizes oxidative damage during the process. However, it is an expensive process that causes significant structural damage, resulting in more porous dehydrated products that are more susceptible to quality degradation during storage.

Therefore, this study aims to assess the impact of dehydration (by hot air drying at 60 and 70 °C and freeze-drying) and storage at room temperature and accelerated conditions on the functional quality and stability of almond bagasse powder. Specifically, it seeks to explore the effects of dehydration conditions, temperature, and storage time on the humidity, water activity, higroscopicity, optical properties, acidity, peroxide index, antiradical capacity, polyphenol profile, and microbiological stability of almond bagasse powder.

## 2. Materials and Method

### 2.1. Process for Obtaining Almond Bagasse and Almond Bagasse Powder

Naturally peeled almonds were purchased from a nearby grocery store and immersed in tap water at a 1:9 ratio by weight for 18 h. The soaked almonds were ground at 10,000 rpm for 20 s using a domestic food processor (Thermomix^®^, Vorwerk, Spain), and subsequently filtered through a 500 µm stainless steel sieve to recover the almond bagasse for further characterization.

To obtain a powder, the almond bagasse was evenly distributed on a plastic grid with a nominal opening of 2 mm and air-dried until the water activity (aw) was below 0.3.

A convective dryer (Pol-eko Aparatura, Katowice, Poland) with cross-flow air at a velocity of 10 m/s at 60 or 70 °C for 10 h and 7 h, respectively, was used to obtain air-dried (HAD) bagasse. The air temperature was chosen to minimize damage to macro- and micronutrients and to keep the characteristics of the dehydrated bagasse as close as possible to those of the raw material.

The samples were subjected to freezing at −40 °C for 24 h in a CVN-40/105 freezer (Matek, Barcelona, Spain), followed by sublimation at −45 °C (condenser temperature) and 0.1 mbar for 48 h in a LyoQuest-55 laboratory freeze drier (Telstar, Terrasa, Spain).

The dried almond bagasse, HAD and LYO, was ground in a food processor (Thermomix^®^, Vorwerk, Spain) in two stages: first at 4000 rpm for 20 s at 5 s intervals, then at 10,000 rpm for 20 s at 5 s intervals to obtain a coarse-grained powder.

### 2.2. Evaluation of Storage Stability of the Almond Bagasse Powder

The almond bagasse powders, dried at 60 and 70 °C and lyophilized, were packed in heat-sealed, aluminum-laminated polyethylene bags (200 g) (ASIN B0C4CLL6JH, Amazon, Spain) and stored at room temperature (27 ± 2 °C, 50–75% RH) and under accelerated conditions (38 ± 2 °C, 90 ± 2% RH) for 180 days. Water activity, moisture, optical properties, hygroscopicity, acidity, total phenols, antiradical activity, and peroxides index were determined every 15 days, and specific phenolic compounds were determined every 90 days. Microbiological analyses (molds, yeasts, and total mesophiles) were performed monthly. Each analysis was performed in triplicate.

### 2.3. Physico-Chemical Properties

The water activity of almond bagasse powders (HAD60, HAD70 and LYO) was determined at 20 °C with a dew point hygrometer (DECAGÓN Aqualab 4TE).

The moisture content was determined following the official method established by the AOAC in dried fruits [18].

Hygroscopicity was evaluated according to the method described by [19]; 0.5 g of each sample was weighted into a glass crucible and placed into a closed chamber with saturated sodium sulfate solution (Na_2_SO_4_) (Sigma-Aldrich, St. Louis, MO, USA) for 7 days at 25 °C. After 7 days, the weight gain was measured.

The CIE**L***a***b** coordinates were determined with a spectrocolorimeter (MINOLTA, CM-3600D, Osaka, Japan) using the standard light source D65, the standard 10° observer, and the surface reflectance spectra from 400 to 700 nm. The hue (h_*ab*_), chroma (C_*ab*_), and color difference (Δ*E*) were calculated by Equations (1)–(3).
(1)$$h_{ab}=\mathrm{arctg}\left(\frac{b}{a}\right)$$
(2)$$C_{ab}=\sqrt{a^{2}+b^{2}}$$
(3)$$\Delta E=\sqrt{\Delta L^{2}+\Delta a^{2}+\Delta b^{2}}$$
where *a**, *b**, and *L** represent the color coordinates in the CIE color space, *L** stands for lightness, *a** is the red-green component, and *b** is the yellow-blue component. h_*ab*_, C_*ab*_, and Δ*E* refer to the hue, chroma, and color difference of each powder after storage for a period of time t compared to the powder before storage, respectively.

### 2.4. Acidity and Peroxide Value

The standardized method proposed by Rani et al. [20] was employed to determine the acidity. Following the pH measurement, the sample was neutralized with 0.5 M NaOH (Sigma-Aldrich, St. Louis, MO, USA) until it reached a pH of 8.20. Results were expressed as g oleic acid/100 g dry matter.

The peroxide index was determined in accordance with the official AOAC method [21] for the peroxides values of oils and fats. Then, 0.5 g of the sample was weighed and solubilized in an aqueous solution containing 18 mL of acetic acid and 12 mL of graphyl chloroform (Sigma-Aldrich, St. Louis, MO, USA). After that, 0.5 mL of a saturated solution of KI (Sigma-Aldrich, St. Louis, MO, USA) and 30 mL of distilled water were added and titrated with a 0.01 M Na_2_S_2_O_3_ solution (Sigma-Aldrich). Results were expressed as meq O_2_/kg dry matter.

### 2.5. Determination of Phenolic Compounds and Antiradical Capacity

For the extraction of antioxidants, 1 g of sample was weighed and mixed with 10 mL of a mixture of methanol and water in a ratio of 80:20 (*v*/*v*) as a solvent solution. After 1 h of shaking (Intelli-Mixer RM-2M), the sample was centrifuged (Selecta, “Medrifriger BL-S”) at 10,000× rpm and 20 °C for 5 min. Determinations were carried out on the supernatant, hereafter called the extract.

#### 2.5.1. Total Phenol Content

The quantification of total phenols was conducted following the method outlined by Wolfe et al. [22]. In a spectrophotometric cuvette, 0.125 mL of Folin–Ciocalteu reagent (Sigma-Aldrich, St. Louis, MO, USA), 0.125 mL of extract, and 0.5 mL of double-distilled water were combined and allowed to react for 6 min. Subsequently, 1.25 mL of 7% (*w*/*v*) sodium carbonate (Sigma-Aldrich, St. Louis, MO, USA) solution and 1 mL distilled water were added. A reagent blank was made; the extract was replaced with double-distilled water and allowed to react for 90 min. A blank reagent was prepared by substituting the extract with double-distilled water. The absorbance was then measured at 765 nm using a spectrophotometer (Thermo Scientific, Helios Zeta U/Vis, Leicestershire, UK). The results obtained were compared with a gallic acid standard curve (Purity of ≥98%, Sigma-Aldrich, St. Louis, MO, USA) and presented as milligrams of gallic acid equivalents per gram of dry matter (mg GAE/g dm).

#### 2.5.2. Antiradical Capacity by DPPH and ABTS Methods

The antioxidant capacity was determined following the DPPH procedure described by Stratil et al. [23] and Kuskoski et al. [24], with some modifications. A total of 2.9 mL of the methanol–DPPH solution (39.4 mg/mL concentration) and 0.1 mL of the extract were mixed in a spectrophotometric cuvette. The mixture was left to react for 60 min, and the absorbance was measured at 517 nm using a spectrophotometer (Thermo Scientific, Helios Zeta U/Vis, Leicestershire, UK). The results obtained were compared with a calibration curve of Trolox (C14H18O4, purity ≥ 97%, Sigma-Aldrich, St. Louis, MO, USA) and expressed as milligram Trolox equivalents per gram of dry matter (mg TE/g dm).

Moreover, antiradical activity was assessed by means of the ABTS (2,2-azino-bis-3-ethylbenzothiazolin-6-sulphonic acid) method, following the procedure described by Re et al. [25]. An acidic solution (7 mM, ≥99% purity) was prepared together with potassium persulphate (2.45 mM, 99% purity) in distilled water, which was incubated in the dark overnight. Subsequently, the absorbance was adjusted to 1.0 ± 0.02 using methanol at 734 nm. For the control, the sample was replaced with distilled water. Samples were measured at 0, 3, and 7 min at a wavelength of 734 nm in a spectrophotometer (Thermo Scientific, Helios Zeta UV/Vis, Leicestershire, UK). Results were expressed as milligrams of trolox equivalent per gram of dry matter (mg TE/g dm).

#### 2.5.3. Phenolic Compounds by HPLC Analysis

Specific polyphenolic compounds were determined following the method proposed by Caprioli et al. [26] and Giusti et al. [27]. For acid hydrolysis, 2.5 g of the sample was taken and mixed with 7.5 mL of the solvent (70:30 *v*/*v* ethanol and distilled water). The pH was adjusted to 4 with HCL and left for 2 h in an ultrasonic bath at room temperature. After that, the samples were centrifuged at 8000× *g* for 15 min. This extraction process was performed twice on the solid sample. The supernatants were filtered with 20 µm polytetrafluoroethylene (PTFE), and the resulting extract was analyzed by HPLC.

Samples extracts were analyzed using a 1200 series rapid resolution HPLC coupled to a series diode array detector (Agilent, Palo Alto, CA, USA). Phenolic compounds were separated on a Brisa-LC 5 column [28]. Mobile phase A was 1% formic acid, and mobile phase B was acetonitrile. The gradients were: 0 min, 10% B; 25 min, 60% B; 26 min, 80% B, with holding up to 30 min; and 35 min, 10% B, with holding up to 40 min. The working conditions were an injection volume of 10 µL and a flow rate of 0.5 mL/min at 30 °C. The phenolic compounds were detected at different chromatographic retention times determined by reference standards. The wavelengths of each compound were: vanillin, 250 nm; 260 nm for rutin, 4-hydroxybenzoic acid, and quercetin 3-glucoside; 280 nm for chlorogenic acid and epicatechin; and 320 nm for sinapic acid, ferulic acid, p-coumaric acid, and 7-glucoside of apigenin. The compounds were quantified by calibration curves and the results were expressed as mg/100 g dm.

### 2.6. Microbiological Analyses

A 10 g sample of each almond bagasse powder was aseptically transferred into sterile stomacher bags. Samples were mixed with 90 mL of sterile peptone solution (Scharlab, Barcelona, Spain) and homogenized for 2 min. Serial dilutions were made for plate inoculation in 9 mL of sterile peptone. Plate count agar (Scharlab, Barcelona, Spain) was used for total plate count and potato dextrose agar (Scharlab, Barcelona, Spain) for the yeast and mold count. Total mesophiles plate count, yeasts, and molds were incubated at 37 ± 2 °C for 24 h and at 27 ± 2 °C for 72 h, respectively.

### 2.7. Statistical Analysis

The results underwent statistical analysis using Statgraphics Centurion XVI.I software (Statpoint Technologies, Inc., Warrenton, VA, USA) at a 95% confidence level (*p*-value < 0.05). The normality of the data was assessed using the Shapiro–Wilk test (*p* > 0.05). Following this, an analysis of variance (ANOVA) was conducted. Each treatment transformation was replicated in three separate experiments, with three replicates in each experiment. Fisher’s LSD test was utilized to identify any significant differences (*p*-value < 0.05) among groups.

## 3. Results and Discussion

Table 1 shows the physicochemical properties of almond bagasse after air drying at 60 °C and 70 °C and freeze-drying. The difference in water activity (aw) among the hot air-dried samples at different temperatures and the freeze-dried ones (samples HAD60, HAD70, and LYO) is striking. This may be due to the structural fracture produced during the freezing and sublimation stages. This structural rupture facilitates the removal of more bound water, allowing for a greater reduction in water activity. In hot-air-dried samples, the drying conditions caused phase transitions and a compact and deformed structure, which made water removal more difficult. In all cases, the water activity was reduced to below 0.3, which is the recommended threshold for ensuring microbiological and physicochemical stability in powdered products such as milk powder or instant coffee [29,30]. Nevertheless, as the storage period progressed, there was an observed increase in water activity for all samples (Table 2 and Table 3). This increase was most pronounced for samples subjected to accelerated storage, with the values for samples HAD60 AC, HADO70 AC, and LYO AC increasing by 3, 2, and 8 times their initial values, respectively. The higher porosity associated with the structural damage suffered by the freeze-dried samples justifies their higher water uptake capacity. However, while a substantial increase in water activity was noted at the end of the storage period, it did not reach critical levels in any case.

The moisture content in the samples was low, and there were no significant differences among the various drying methods employed. However, the rate of water absorption in the samples during storage was noted to be higher under accelerated conditions. This absorption significantly increased, reaching values up to 3, 4, and 25 times higher than their initial levels for the HAD60 AC, HAD70 AC, and LYO AC samples, respectively. However, even though there was moisture absorption by the end of the storage period (180 days), the moisture content remained below 10%, which is still considered low.

Hygroscopicity is the capacity of a material or powder to absorb moisture and come into equilibrium with the relative humidity of the environment. As per Arlindo et al. [31], the hygroscopic properties of specific foods largely depend on their chemical compositions and storage conditions, including relative humidity. Significant differences were observed among air-dried samples at 60 °C and 70 °C and lyophilized ones, the values of the latter being higher. As a consequence, the lyophilized almond powders showed increased humidity and water activity after storage, as has been remarked previously. The hygroscopic values can be considered low to medium, as a material can be considered non-hygroscopic if less than 20% is observed [32]. This could elucidate the increased moisture absorption under accelerated conditions. Comparable results have been documented for guava pulp powder and melon powder, both stored under normal and accelerated conditions [33,34].

The acidity of food is attributed to the presence of organic acids that serve as substrates for respiration, and variations in this parameter can impact the quality characteristics of the food [35]. The acidity (Figure 1) showed significant differences (*p*-value < 0.05) between air-dried samples and freeze-dried ones, and significant differences (*p*-value < 0.05) were observed among the HAD60, HAD70, and LYO samples in terms of storage at different conditions. The samples stored for 180 days at room temperature presented values ranging from 0.56 to 0.97 g/100 g dry matter, with the lowest acidity observed in the freeze-dried samples and the highest in the samples dried at 70 °C. The samples subjected to accelerated storage for 180 days exhibited a significant increase compared to those stored at room temperature; there were increases of 73%, 61%, and 25% for the HAD60 AC, HAD70 AC, and LYO AC samples, respectively. According to Morrison [36], the increase in acidity in stored grains and flours over an extended period is due to an elevation in the concentration of free fatty acids, which leads to greater food deterioration. As stated by [37,38], other potential factors contributing to the increased acidity in samples stored at higher temperatures include the interaction of the amino groups in amino acids, short-chain peptides, and proteins, resulting in the release of carboxylic ends while generating acidic by-products from advanced Maillard reactions. Considering that almond bagasse powder is rich in fatty acids and proteins, the observed increases could be attributed to both phenomena, especially in the hot-air-dried samples. However, even with accelerated storage, the values obtained after 180 days of storage are acceptable, taking into account that, in foods such as olive oil, values of up to 1 g oleic acid/100 g can be obtained without affecting quality [39].

The peroxides index values for air-dried samples at 60 and 70 °C and freeze-dried ones ranged from 1.9 to 3.4 meq O_2_/kg dry matter (Figure 1). Significant differences (*p*-value < 0.05) were observed between treatments, as the hot-air-drying process is more aggressive in terms of oxidative reactions due to the presence of oxygen and the high temperature, resulting in a higher peroxide index [40]. Samples stored for 180 days showed peroxide values ranging from 10 to 12.8 meq O_2_/kg dry matter for samples stored at room temperature and from 14.7 to 23 meq O_2_/kg dry matter for samples subjected to accelerated storage. The peroxides index values were higher in accelerated storage for the different drying methods. Similar results were found by El Bernoussi et al. [41] in almond oils subjected to accelerated storage for 4 weeks, with peroxides indices ranging from 2.4 to 24.6 O_2_/kg. As the temperature increased, oleic, linoleic, and linolenic acids produced hydroperoxides. These substances decomposed, resulting in the creation of a broad spectrum of secondary oxidation products. Hydroperoxides were primarily formed and decomposed during the initial stages. Nevertheless, as the storage period progressed, the rate of formation substantially rose, leading to a significant elevation in the overall concentration of hydroperoxides. Consequently, this heightened concentration contributed to increased oxidation. The oxidation rate escalated exponentially with the temperature. Additionally, there was also an interaction between oxygen and temperature, as higher temperatures lead to decreased reduced oxygen availability. Similar trends were observed in dried hazelnuts stored for 21 months, showing an increase of up to twice their initial value [42]. Likewise, comparable results were obtained in pistachios stored for 15 months, recording values of 14.9 meq O_2_/kg dry matter [43] and in milk powder stored for 9 months, where values of 11.3 meq O_2_/kg matter were obtained.

Regarding microbiological stability, the dehydration process significantly reduced the available water for microbial growth, resulting in null values for the final microbial load. Although no significant differences were noted in cell counts between drying methods (*p*-value > 0.05), there was a gradual increase in these values as the storage period extended. Following 180 days of storage, samples subjected to an accelerated storage process exhibited higher levels of mesophiles, molds, and yeasts. Nevertheless, it is essential to highlight that these values did not surpass critical levels, so all samples met the minimum safety requirements in accordance with the Spanish Government’s Safety and Nutrition Law [44].

The dehydration treatment significantly affected the content of specific polyphenols (*p*-value < 0.05), although the effect depended on the chemical nature of the considered component (see Table 1). The samples subjected to hot air drying (HAD60 and HAD70) showed reduced polyphenol content compared to the freeze-dried samples, except for rutin, which demonstrated values of 206.4 ± 0.9 and 195.2 ± 0.2 for samples HAD60 and HAD70, respectively, and apigenin-7-glucoside, with values of 60 ± 11 and 65 ± 10 in samples HAD60 and HAD70, respectively. Once more, the moderate temperature and the absence of oxygen during freeze-drying minimized the potential for degradation. In addition, Maillard reaction products, generated during the air-drying treatment, could chelate flavor compounds, transition metal ions, and polyphenols, reducing the possibility of being adequately quantified [45]. However, freezing and sublimation did lead to the degradation of the cellular structure, promoting the formation of a more porous matrix structure. This circumstance could, over time, facilitate the loss of some phenolic compounds, as in the case of rutin and apigenin-7-glucoside. Understanding the reasons why temperature (in the case of air-drying) or structural degradation (in the case of freeze-drying) have a greater influence on the degradation of certain components would require further work taking into account the release of polyphenols in each treatment; the kinetics of degradation reactions associated with each compound; and other compound interactions. The temperature during the hot-air-drying process had a significant impact on the total phenol content (*p*-value < 0.05). In addition, in most cases, drying at 60 °C led to lower phenol content compared to drying at 70 °C. These temperature differences could play a decisive role in the inactivation of enzymes involved in specific degradation reactions. Storage also had a significant impact on the specific polyphenol content (see Table 2 and Table 3). An increase was observed in all samples, but samples subjected to accelerated storage showed an even greater increase. Specifically, for 4-hydroxybenzoic acid, epicatechin, and vanillin, increases of 8%, 63%, and 59%, respectively, were recorded for the HAD60 AC samples; 29%, 90%, and 126%, respectively, for the HAD70 AC samples; and 14%, 13%, and 12%, respectively, for the LYO AC samples. Storage might induce physical transformations in components like cellulose or lignin, thereby facilitating access to the phenolic profile for extraction. Another potential explanation is the continuation of polyphenol synthesis after almonds are harvested, a phenomenon observed in some food items. For instance, in peanuts, there was a 42% increase in polyphenols after 24 months of storage [46]. A similar trend was noted in sprouted groundnuts, which displayed an elevated content of total phenols. Likewise, a comparable pattern emerged in almond shells stored at 23 °C for 9 months, indicating an 84% rise in catechin, a 50% increase in epicatechin, and an 18-fold increase in hydroxybenzoic acid compared to the initial levels [12].

The rise in total phenol content after 180 days of storage manifested prominently in the total polyphenolic profile, particularly in the samples subjected to accelerated storage. Notably, HAD60 AC, HAD70 AC, and LYO AC exhibited increases of 25%, 52%, and 7%, respectively. A substantial elevation was observed in the HAD70 AC samples, potentially attributed to the utilization of high temperatures. These elevated temperatures might have induced the generation of Maillard reaction by-products, renowned for their exceptional antioxidant capacity [47]. Nevertheless, previous studies indicated that storing nuts for extended periods results in a significant decrease in the levels of polyphenols and antioxidant compounds. For instance, the storage of peanuts at temperatures of 20 °C and 35 °C for up to 4 months revealed a 35% reduction in total polyphenols compared to the initial levels [48]. Similarly, a 42% decrease in green tea catechins was noted following storage at 20 °C for 4 months [49]. Also, raw almonds stored in darkness for up to 24 months lost between 83 and 90% of their vitamin E content [49].

Table 4 shows the values of antiradical capacity determined by the DPPH and ABTS methods and the content of total phenols. Regarding the effect of the dehydration method, it was observed that the antiradical capacity measured by DPPH showed no significant differences between the samples. However, concerning the antiradical capacity assessed by ABTS, the samples dried by hot air exhibited higher values compared to the freeze-dried ones. This could be attributed to the products formed through Maillard reactions induced by temperatures applied in the hot-air-drying process. These reactions involve amino acids and generate lipid oxidation products. According to Majid et al. [47], the resulting products from Maillard reactions possess excellent antiradical capacities. The differences observed in the results obtained from the DPPH and ABTS methods stemmed from their distinct sensitivities to antiradical compounds. The ABTS radical was found to interact with a greater number of antioxidant compounds. Its shorter reaction time and more hydrophilic nature enabled it to react in both organic and aqueous environments [50]. The total phenols exhibited similar behavior, with hot-air-dried samples showing higher values than the freeze-dried ones. However, notably, the samples dried at 70 °C displayed even higher values. This could be pivotal in terms of other degradative enzyme inactivation.

Contrarily, the changes in antiradical capacity during the storage of hot-air-dried and freeze-dried almond bagasse powders showed a gradual increase in all dehydrated samples over time. Higher values were noted in the samples subjected to accelerated storage. With the DPPH method, there was an increase of 0.085 mg TE/g dry matter for the LYO AC samples, while the HAD60 samples showed an increase of 0.07 mg TE/g of dry matter. Regarding the ABTS method, increases of 0.03 and 0.04 mg TE/g of dry matter, respectively, were observed in the HAD60 AC and LYO AC samples. Similar findings were reported by Alessandra et al. [51], who observed an augmentation in antiradical capacity over time in minimally processed citrus segments and juices. This phenomenon could be attributed to the ongoing formation of new compounds with antioxidant properties, such as Maillard reaction products, which persist even after extended storage periods. Throughout the storage period, a noticeable increase in total phenols was observed, particularly evident in samples undergoing accelerated storage conditions. Specifically, the HAD60 AC and HAD70 AC samples at 6 months of storage exhibited increases of 0.32% and 55%, respectively. These findings align with those reported by Bolling et al. [12], who similarly observed an augmentation in phenolic compounds within almond skins under comparable storage conditions. However, the LYO AC sample experienced a 21% loss in total phenols. This phenomenon could be related to the freeze-drying process, which, despite occurring at low temperatures and under vacuum conditions, produces an important structural rupture that favors the diffusion of oxygen and the oxidation of phenolic compounds during storage.

In summary, for specific polyphenolic compounds, temperature and oxygen had a significant degradative effect during processing and storage, causing a significant decrease in the content of all the components analyzed, except for rutin, apigenin-7-glucoside, and vanillin. However, the effects were more pronounced with hot air at 60 °C than at 70 °C. On the other hand, the results followed a very different evolution when referring to non-specific measures such as antiradical activity or total phenolic content. In these cases, it seems that lipid oxidation and Maillard reactions, which occurred mainly during hot air drying (and to a greater extent at 70 °C) and continued during storage, generated reaction products with a high antiradical capacity and phenolic nature [45].

The different dehydration methods provided almond bagasse powders with significantly different color coordinates (see Table 1). This difference arose from browning and oxidation reactions, resulting in more reddish and yellowish hues in the samples air-dried at 70 °C (HAD70), as indicated by higher values in the *a** and *b** coordinates. In contrast, the freeze-dried samples displayed greater lightness (*L*) due to the reduced browning caused by the moderate temperature and the absence of oxygen during the process. At the end of the 180-day storage period, it was observed that the samples HA60 and LYO practically did not exhibit significant differences in terms of color variation compared to the initial samples (see Δ*E* in Table 3). However, in the case of sample HAD70, a slight increase in color variation was noted (5.0 ± 0.4), and this trend was consistent in HAD70 AC. Notably, it should be highlighted that color differences are imperceptible to the human eye when Δ*E* is less than 1, but become visibly apparent when the Δ*E* value exceeds 3 [52]. Additionally, it can be observed that the color variation during storage (Table 2 and Table 3) slightly increased in the samples stored at room temperature, and that more intense changes occurred in the samples stored under accelerated conditions. These results are consistent with the fact that Maillard reactions continue during storage and become more intense as the temperature increases.

## 4. Conclusions

Both hot air drying and freeze-drying provided powders with water activity and hygroscopicity values that ensured stability for three months, even under accelerated storage conditions. Moreover, the increases in the peroxides index and acidity during storage at ambient temperature was sufficiently moderate to ensure good quality of the final product in all cases.

The effect of storage on the content of specific polyphenols or the antiradical capacity was mainly determined by the structural changes caused by the dehydration operation and the formation of compounds in reactions such as Maillard reactions in hot air drying. In any case, understanding the reasons why temperature or structural changes have a greater influence on the degradation of certain components or the generation of others would require further work taking into account the release of polyphenols in each treatment; the kinetics of degradation reactions associated with each compound; and other interactions between compounds.

The two dehydration methods tested, hot air and freeze-drying, affected the properties studied to different degrees. The recommendation to use one or the other method would largely depend on the desired properties of the final product. However, scaling up the process to industrial levels should consider the costs associated with energy consumption, the investment required in equipment, and the performance of the operation.

## Figures and Tables

**Figure 1 foods-13-00744-f001:**
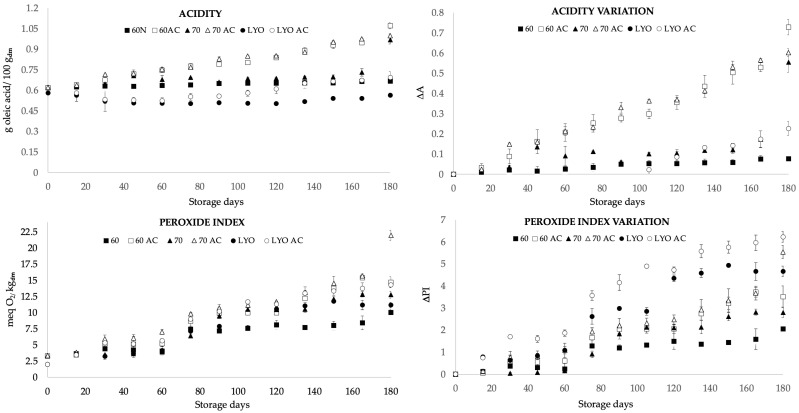
Acidity, peroxide index, acidity variation (ΔA), and peroxide index variation (ΔPI) for almond bagasse powders air-dried at 60 °C (HAD60) and 70 °C (HAD70) and freeze-dried (LYO) during 180 days of storage at room temperature and in accelerate conditions (AC). Variation was calculated as the difference between the value at the time t minus the value at time 0 relative to the value at time 0. Mean ± standard deviation is represented.

**Table 1 foods-13-00744-t001:** Water activity, humidity, higroscopicity, optical properties, peroxides index, acidity, microbial counts, and specific phenolic compounds of air-dried and lyophilized almond bagasse (HAD60: hot-air-dried at 60 °C; HAD70: hot-air-dried at 70 °C; LYO: lyophilized).

	HAD60	HAD70	LYO
Water activity	0.13 ± 0.03 ^b^	0.23 ± 0.01 ^c^	0.05 ± 0.01 ^a^
Humidity (g/g_dm_)	0.009 ± 0.009 ^a^	0.007 ± 0.001 ^a^	0.0011 ± 0.0009 ^a^
Hygroscopicity (g_w_/g)	0.116 ± 0.004 ^a^	0.20 ± 0.01 ^b^	0.224 ± 0.007 ^c^
Peroxide (m_eq_/kg_dm_)	3.3 ± 0.2 ^b^	3.4 ± 0.2 ^b^	1.9 ± 0.1 ^a^
Acidity (g/100 g _dm_)	0.619 ± 0.008 ^b^	0.622 ± 0.001 ^b^	0.58 ± 0.01 ^a^
Mesophiles (log CFU/g_dm_)	n.d	n.d	n.d
Mold and yeast (log CFU/g_dm_)	n.d	n.d	n.d
Phenolic compounds (mg/100 g_dm_)			
4-Hydroxibezoic acid	60.0± 0.9 ^a^	70.7 ± 0.9 ^b^	84.5 ± 1.4 ^c^
Epicatechin	458 ± 17 ^a^	498 ± 7 ^b^	585.0 ± 0.9 ^c^
Vanillin	94.6 ± 0.9 ^a^	128.4 ± 0.9 ^c^	98.7 ± 1.4 ^b^
Rutin	206.4 ± 0.9 ^c^	195.2 ± 0.2 ^b^	151.6 ± 0.9 ^a^
Quercitin 3-glucoside	63.0 ± 0.9 ^a^	66.1 ± 0.9 ^b^	112.2 ± 0.9 ^c^
p-Coumaric acid	39.0 ± 0.9 ^a^	39.4 ± 0.9 ^a^	51.1 ± 0.9 ^b^
Sinapic acid	61.6 ± 0.4 ^a^	63.9 ± 0.4 ^b^	120.5 ± 0.5 ^c^
Ferulic acid	63.1 ± 0.4 ^a^	65.0 ± 0.4 ^b^	107.8 ± 0.5 ^c^
Apigenin-7-glucoside	60 ± 11 ^b^	65 ± 10 ^b^	36 ± 16 ^a^
Clorogenic acid	78.2 ± 0.9 ^a^	81.6 ± 0.9 ^b^	87.4 ± 0.9 ^c^
Sum of specific phenols	1181 ± 12 ^a^	1272 ± 1 ^b^	1435 ± 9 ^c^
Color			
*L*	58.1 ± 0.5 ^b^	51.3 ± 0.5 ^a^	61.86 ± 0.09 ^c^
*a**	7.15 ± 0.06 ^a^	8.1 ± 0.2 ^b^	8.0 ± 0.6 ^b^
*b**	15.27 ± 0.05 ^a^	19.3 ± 0.4 ^c^	16.5 ± 0.4 ^b^
C_*ab*_	64.93 ± 0.02 ^a^	67.2 ± 0.4 ^c^	65.1 ± 0.5 ^b^

Mean ± standard deviation of three repetitions. Different superscript letters in the same line indicate statistically significant differences with a confidence level of 95%. dm, dry matter; aw, water activity; X_w_, water content; n.d, not detected.

**Table 2 foods-13-00744-t002:** Water activity, humidity, higroscopicity, optical properties, peroxides index, acidity, microbial counts, and phenolic compounds for air-dried and lyophilized almond bagasse (HAD60: hot-air-dried at 60 °C; HAD70: hot-air-dried at 70 °C; LYO: lyophilized) after 90 days of storage at room temperature and in accelerated conditions (AC).

	HAD60	HAD70	LYO	HAD60 AC	HAD70 AC	LYO AC
Water activity	0.245 ± 0.001 ^b^	0.274 ± 0.002 ^c^	0.10 ± 0.01 ^a^	0.341 ± 0.001 ^d^	0.41 ± 0.02 ^e^	0.30 ± 0.02 ^c^
X_w_ (g/g_dm_)	0.013 ± 0.01 ^b^	0.0137 ± 0.0002 ^b^	0.0030 ± 0.0001 ^a^	0.017 ± 0.001 ^c^	0.026 ± 0.004 ^d^	0.0183 ± 0.0005 ^c^
Hygroscopicity (g_w_/g)	0.195 ± 0.006 ^a^	0.220 ± 0.004 ^b^	0.22 ± 0.02 ^b^	0.202 ± 0.009 ^c^	0.208 ± 0.005 ^d^	0.208 ± 0.006 ^cd^
Peroxides (m_eq_/g_dm_)	7.14 ± 0.08 ^a^	9.5 ± 0.4 ^c^	7.9 ± 0.1 ^b^	9.92 ± 0.02 ^a^	10.7 ± 0.5 ^b^	10.2 ± 0.7 ^b^
Acidity (g/100 g_dm_)	0.65 ± 0.01 ^c^	0.68 ± 0.01 ^c^	0.52 ± 0.03 ^a^	0.79 ± 0.01 ^d^	0660 ± 0.006 ^e^	0.579 ± 0.004 ^b^
Mesophiles (log CFU/g_dm_)	1 ± 0 ^b^	0 ^a^	1 ± 0 ^b^	1.5 ± 0.7 ^b^	0 ^a^	2 ± 0 ^c^
Mold and yeast (log CFU/g_dm_)	1 ± 0 ^b^	0 ^a^	1 ± 0 ^b^	1.5 ± 0.7 ^a^	1.5 ± 0.7 ^a^	1.5 ± 0.7 ^a^
	Phenolic compounds (mg/100 g_dm_)					
4-Hydroxibezoic acid	69.7 ± 0.5 ^d^	72.1 ± 0.1 ^e^	28.7± 0.3 ^a^	66.7 ± 0.6 ^c^	72.6 ± 0.6 ^e^	34.3 ± 0.5 ^b^
Epicatechin	460.6 ± 0.1 ^a^	773.7 ± 33 ^b^	701.0 ± 38 ^b^	441 ± 86 ^a^	829.9 ± 17 ^c^	697.6 ± 21 ^b^
Vanillin	256.8 ± 4 ^d^	310.9 ± 1 ^f^	87.3 ± 0.9 ^b^	229.8 ± 2 ^c^	294.5 ± 3 ^e^	79.0 ± 0.9 ^a^
Rutin	154.7 ± 0.3 ^b^	153.1 ± 0.8 ^b^	256.3 ± 3 ^e^	162.9 ± 0.5 ^c^	146.7 ± 0.03 ^a^	242.2 ± 1.1 ^d^
Quercitin 3-glucoside	63.0 ± 0.04 ^b^	51.9 ± 0.2 ^a^	106.9 ± 1.2 ^d^	61.4 ± 0.8 ^b^	51.5 ± 1.2 ^a^	82.1 ± 1 ^c^
p-Coumaric acid	36.6 ± 1.1 ^a^	52.9 ± 0.3 ^c^	39.9 ± 0.04 ^b^	59.3 ± 1.3 ^d^	53.1 ± 1.7 ^c^	39.6 ± 0.8 ^b^
Sinapic acid	78.0 ± 0.7 ^b^	87.0 ± 0.9 ^c^	69.9 ± 0.5 ^c^	80.4 ± 0.6 ^b^	85.2 ± 0.3 ^c^	72.6 ± 2 ^a^
Ferulic acid	77.3 ± 0.8 ^b^	83.2 ± 0.8 ^c^	70.9 ± 1 ^a^	79.0 ± 0.8 ^b^	82.5 ± 1 ^c^	72.0 ± 0.6 ^a^
Apigenin-7-glucoside	77 ± 1 ^c^	69.2 ± 0.6 ^a^	102 ± 6 ^d^	75.5 ± 0.1 ^bc^	69.6 ± 0.3 ^ab^	100.8 ± 0.8 ^d^
Clorogenic acid	82.2 ± 3 ^a^	88.7 ± 1 ^b^	98.6 ± 1 ^c^	85.8 ± 1 ^ab^	102.40 ± 0.01 ^c^	99.7 ± 2 ^c^
Sum of specific phenols	1356.4 ± 0.4 ^a^	1743 ± 29 ^c^	1562 ± 25 ^b^	1342 ± 4 ^a^	1788 ± 8 ^c^	1520 ± 3 ^b^
	Color					
*L*	58.6 ± 0.2 ^b^	59.3 ± 0.2 ^c^	61.7 ± 0.6 ^d^	57.22 ± 0.12 ^a^	58.7 ± 0.2 ^b^	72.94 ± 0.01 ^e^
*a**	6.99 ± 0.14 ^c^	6.80 ± 0.04 ^b^	7.06 ± 0.14 ^c^	7.33 ± 0.02 ^d^	7.30 ± 0.01 ^d^	5.29 ± 0.01 ^a^
*b**	15.2 ± 0.5 ^b^	15.72 ± 0.09 ^c^	15.78 ± 0.03 ^c^	15.86 ± 0.12 ^c^	16.55 ± 0.09 ^d^	14.67 ± 0.01 ^a^
C_*ab*_	16.7 ± 0.5 ^b^	17.13 ± 0.07 ^c^	17.29 ± 0.03 ^c^	17.10 ± 0.01 ^c^	18.09 ± 0.08 ^d^	15.60 ± 0.01 ^a^
Δ*E*	0.63 ± 0.44 ^a^	1.3 ± 0.6 ^a^	1.3 ± 0.7 ^a^	1.2 ± 0.1 ^a^	6.6 ± 0.5 ^c^	3.19 ± 0.04 ^d^

Mean ± standard deviation of three repetitions. Different superscript letters in the same line indicate statistically significant differences with a confidence level of 95%. dm, dry matter; aw, water activity; X_w_, water content.

**Table 3 foods-13-00744-t003:** Water activity, humidity, higroscopicity, optical properties, peroxides index, acidity, microbial counts, and specific phenolic compounds for air-dried and lyophilized almond bagasse (HAD60: hot-air-dried at 60 °C; HAD70: hot-air-dried at 70 °C; LYO: lyophilized) after 180 days of storage at room temperature and under accelerated conditions (AC).

	HAD60	HAD70	LYO	HAD60 AC	HAD70 AC	LYO AC
Water activity	0.30 ± 0.03 ^ab^	0.34 ± 0.03 ^b^	0.280 ± 0.002 ^a^	0.431 ± 0.002 ^a^	0.484 ± 0.002 ^b^	0.432 ± 0.004 ^a^
Humidity (g/g_dm_)	0.021 ± 0.001 ^b^	0.024 ± 0.001 ^b^	0.0154 ± 0.0004 ^a^	0.0314 ± 0.0004 ^d^	0.032 ± 0.002 ^d^	0.0292 ± 0.0005 ^c^
Hygroscopicity (g_w_/g)	0.168 ± 0.006 ^a^	0.214 ± 0.005 ^b^	0.213 ± 0.005 ^b^	0.179 ± 0.01 ^a^	0.220 ± 0.013 ^b^	0.223 ± 0.006 ^b^
Peroxide (m_eq_/g_dm_)	10.0 ± 0.6 ^a^	12.8 ± 0.5 ^c^	11.2 ± 0.4 ^b^	14.7 ± 0.9 ^d^	23 ± 2 ^e^	14.3 ± 0.3 ^d^
Acidity (g/100 g _dm_)	0.666 ± 0.004 ^b^	0.97 ± 0.03 ^d^	0.56 ± 0.01 ^a^	1.07 ± 0.02 ^e^	1.00 ± 0.01 ^e^	0.734 ± 0.009 ^c^
Mesophiles (log CFU/g_dm_)	2.5 ± 0.7 ^b^	1 ± 0 ^a^	2.5 ± 0.7 ^b^	3 ± 0 ^c^	1.5 ± 0.7 ^a^	3 ± 0 ^c^
Mold and yeast (log CFU/g_dm_)	2.5 ± 0.7 ^c^	1 ± 0 ^a^	2 ± 0 ^b^	4 ± 0 ^b^	3 ± 0 ^a^	4 ± 1 ^c^
	Phenolic compounds (mg/100 g_dm_)					
4-Hydroxibezoic acid	80.8 ± 1.3 ^c^	75.6 ± 0.2 ^b^	89 ± 2 ^d^	64.6 ± 0.4 ^a^	91.0 ± 0.4 ^e^	96 ± 4 ^f^
Epicatechin	463 ± 7 ^a^	747 ± 6 ^c^	607 ± 4 ^b^	747.5 ± 6.1 ^c^	945.6 ± 24.7 ^d^	659.1 ± 0.9 ^a^
Vanillin	263 ± 6 ^c^	322 ± 2 ^e^	113 ± 1 ^a^	150 ± 2 ^b^	290.3 ± 0.5 ^d^	110.7 ± 1.2 ^a^
Rutin	143.6 ± 0.9 ^c^	145± 3 ^c^	75 ± 2 ^a^	111 ± 2.0 ^b^	173.4 ± 0.4 ^d^	226.1 ± 0.3 ^e^
Quercitin 3-glucoside	68 ± 2 ^c^	61.4 ± 0.4 ^b^	81 ± 2 ^c^	53.2 ± 1.1 ^a^	56.4 ± 0.6 ^a^	76.9 ± 1.4 ^d^
p-Coumaric acid	55 ± 2 ^d^	38 ± 1 ^a^	48.5 ± 0.2 ^c^	53.4 ± 0.4 ^d^	46.3 ± 0.7 ^b^	49.1 ± 0.2 ^c^
Sinapic acid	80 ± 2 ^b^	91.1 ± 1.5 ^c^	76.2 ± 0.4 ^a^	71.4 ± 1.2 ^a^	84 ± 2 ^b^	74 ± 2 ^a^
Ferulic acid	79 ± 2 ^b^	86.9 ± 0.8 ^c^	74.0 ± 1.2 ^a^	71.5 ± 0.1 ^a^	84.7 ± 1.1 ^c^	72.9 ± 0.3 ^a^
Apigenin-7-glucoside	80.6 ± 0.1 ^c^	75 ± 2 ^bc^	81 ± 6 ^d^	68.3 ± 1.1 ^a^	71.6 ± 1.4 ^b^	79 ± 8 ^c^
Clorogenic acid	75 ± 2 ^a^	87.7 ± 0.9 ^b^	88.0 ± 1.4 ^b^	83.2 ± 0.6 ^b^	95.1 ± 1.2 ^d^	91.0 ± 0.4 ^c^
Sum of specific phenols	1388 ± 10 ^b^	1923 ± 1 ^e^	1336 ± 1 ^a^	1475 ± 9 ^c^	1938 ± 4 ^f^	1541 ± 2 ^d^
	Color					
*L*	57.883 ± 0.004 ^c^	54.5 ± 0.3 ^a^	60.20 ± 0.01 ^d^	56.593 ± 0.006 ^b^	56.73 ± 0.01 ^bc^	61.587 ± 0.002 ^e^
*a**	6.85 ± 0.01 ^a^	8.98 ± 0.13 ^d^	6.958 ± 0.002 ^a^	9.00 ± 0.01 ^e^	7.0 ± 0.6 ^b^	7.664 ± 0.002 ^c^
*b**	14.84 ± 0.01 ^a^	18.1 ± 0.4 ^a^	15.25 ± 0.01 ^a^	19.09 ± 0.02 ^c^	14.93 ± 0.01 ^a^	16.643 ± 0.004 ^b^
C_*ab*_	65.22 ± 0.01 ^e^	63.6 ± 0.3 ^c^	65.47 ± 0.01 ^c^	64.74 ± 0.02 ^d^	65.972 ± 0.003 ^f^	65.274 ± 0.003 ^b^
Δ*E*	0.7 ± 0.3 ^a^	5.0 ± 0.4 ^e^	2.3 ± 0.4 ^b^	4.5 ± 0.2 ^d^	7.1 ± 0.6 ^f^	3.56 ± 0.01 ^c^

Mean ± standard deviation of three repetitions. Different superscript letters in the same line indicate statistically significant differences with a confidence level of 95%. dm, dry matter.

**Table 4 foods-13-00744-t004:** Total phenol content and antiradical capacity by DPPH and ABTS methods for hot-air-dried (HAD60: hot-air-dried at 60 °C; HAD70: hot-air-dried at 70 °C) and freeze-dried (LYO) almond bagasse powders during 180 days of storage at room temperature and in accelerate conditions (AC).

	Days	HAD60	HAD70	LYO	HAD60 AC	HAD70 AC	LYO AC
DPPH (mgTE/g _dm_)	0	0.20 ± 0.02 ^a^	0.23 ± 0.01 ^a^	0.24 ± 0.007 ^a^	0.20 ± 0.02 ^a^	0.23 ± 0.01 ^b^	0.319 ± 0.007 ^c^
	60	0.201 ± 0.013 ^a^	0.218 ± 0.02 ^ab^	0.22 ± 0.02 ^c^	0.2372 ± 0.0013 ^b^	0.21 ± 0.01 ^a^	0.333 ± 0.009 ^d^
	90	0.391 ± 0.006 ^d^	0.37 ± 0.01 ^a^	0.36 ± 0.01 ^bc^	0.37 ± 0.01 ^bcd^	0.34 ± 0.01 ^b^	0.38 ± 0.01 ^cd^
	180	0.31 ± 0.02 ^a^	0.307 ± 0.014 ^c^	0.344 ± 0.009 ^d^	0.27 ± 0.01 ^b^	0.213 ± 0.009 ^a^	0.404 ± 0.007 ^e^
ABTS (mgTE/g _dm_)	0	1.04 ± 0.02 ^b^	1.092 ± 0.014 ^b^	0.56 ± 0.05 ^a^	1.04 ± 0.02 ^b^	1.092 ± 0.014 ^b^	0.56 ± 0.05 ^a^
	60	0.73 ± 0.02 ^b^	0.788 ± 0.012 ^c^	0.53 ± 0.02 ^a^	0.75 ± 0.08 ^ab^	0.84 ± 0.04 ^b^	0.65 ± 0.04 ^a^
	90	0.88 ± 0.05 ^c^	0.95 ± 0.04 ^d^	0.44 ± 0.01 ^a^	0.86 ± 0.03 ^c^	0.91 ± 0.04 ^cd^	0.52 ± 0.02 ^b^
	180	0.79 ± 0.05 ^c^	0.99 ± 0.03 ^d^	0.55 ± 0.02 ^a^	1.07 ± 0.01 ^e^	0.813 ± 0.003 ^c^	0.60 ± 0.02 ^b^
Phenols (mg GAE/g _dms_)	0	0.31 ± 0.02 ^a^	0.355 ± 0.006 ^b^	0.313 ± 0.007 ^a^	0.31 ± 0.02 ^a^	0.355 ± 0.006 ^b^	0.313 ± 0.007 ^a^
	60	0.39 ± 0.02 ^b^	0.535± 0.015 ^cd^	0.291 ± 0.004 ^a^	0.47 ± 0.02 ^c^	0.54 ± 0.02 ^e^	0.3 ± 0.4 ^a^
	90	0.323 ± 0.002 ^a^	0.37 ± 0.01 ^b^	0.32 ± 0.01 ^a^	0.44 ± 0.03 ^c^	0.43 ± 0.02 ^c^	0.34 ± 0.01 ^ab^
	180	0.443 ± 0.011 ^b^	0.51 ± 0.02 ^c^	0.242 ± 0.006 ^a^	0.63 ± 0.06 ^d^	0.552 ± 0.011 ^c^	0.248 ± 0.013 ^a^

Mean ± standard deviation of three repetitions. Different superscript letters for the same determination indicate statistically significant differences, with a confidence level of 95%. dm, dry matter; GAE, acid gallic equivalents; TE, Trolox equivalent.

## Data Availability

The original contributions presented in the study are included in the article, further inquiries can be directed to the corresponding author.

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
