# Peer review of "Shelf Life and Functional Quality of Almond Bagasse Powders as Influenced by Dehydration and Storing Conditions"

_foods, 2024, doi:10.3390/foods13050744_

Round 1

Reviewer 1 Report

Comments and Suggestions for Authors

Bagasse produced during almond processing technologies is rich in valuable compounds, but preservation processes are needed to minimize the loss of these components. Dehydration technologies, operated by conventional hot air drying or freeze drying, can be considered suitable for this purpose. Detailed analysis of the effects of drying parameters on microbial state and safety, valuable component loss (complex quality), and furthermore, energy efficiency and economy of the processes, can provide useful data and information for industry practice as well. As the authors established, storage conditions also have significant effects on the shelf life and quality of the dehydrated product. Therefore, the topic of the manuscript foods-2876242 can be considered relevant. The manuscript has a logical structure. The introduction summarizes the relevance of the study well. The applied analytical methods are adequate for the specifics/characteristics of the samples and the main aims of the research. The manuscript contains valuable results that are discussed in detail with relevant references but need revision before publishing.

Comments and Suggestions:

  1. Please clearly define the novelties of the study and summarize them in 1-2 sentences in the Introduction section.
  2. Please explain and give in the manuscript how the temperature range for hot air drying was selected.
  3. There is missing information in the methodology part (air velocity/flow rate and air humidity in hot air drying; applied temperature, time, pressure in freeze drying process, etc.). Please amend these subsections with relevant data and information.
  4. In line 221, authors mentioned that '…the higher drying temperature causes phase transition…'. Please explain this effect (in my opinion, it is strongly affected by the temperature, but the temperature range in the present research is not high).
  5. Please briefly discuss the industry-scale applicability and economy of the applied drying methods as well.
  6. Please reconsider the presentation of experimental data. The manuscript contains lot of experimental data but just one figure.
  7. The visibility of Figure 1 is very poor (mainly axis titles, scales, units). Please improve.
  8. The manuscript contains many typing and grammatical errors. Please check and correct them.
  9. Please unify the reference styles."

Author Response

The authors would like to thank reviewer 1 for the efforts in reviewing the paper and welcome comments and suggestions, which will certainly improve the quality of the paper.

Your suggestions have been answered one per one in the attached file.

Reviewer 2 Report

Comments and Suggestions for Authors

This research investigated the potential use of almond bagasse as a functional ingredient, focusing on the material's stability after dehydration and storage processes. However, some shortcomings and areas for improvement can be identified:

- Remove the punctuation (.) at the end of the Title.

 Abstract:

 - Although it has been mentioned that almond bagasse is prone to lipid oxidation due to its high fat content, it would be helpful to include measurements of lipid oxidation.

- The research mainly focused on preserving antioxidant compounds during the dehydration process and storage. It would be beneficial to include antioxidant activity results to determine the impact of different dehydration techniques and storage on the functional properties of the final product.

- Complex sentence: The phrase "These compounds exhibited an increase at the end of the storage period, particularly noticeable in the samples treated with hot air" could be simplified to improve readability. For example, it could be rewritten as "These compounds increased over the storage period, especially in the samples treated with hot air."

- Overall, the writing and grammar are good, but minor adjustments can be made to improve clarity and readability.

Introduction:

- The introduction addresses a variety of topics related to the nutritional properties of almonds, the production of almond-based beverages, and the potential use of almond bagasse as a functional ingredient. It would be helpful to organize these information into distinct paragraphs to facilitate understanding and clearly highlight the main points.

- Some paragraphs seem disjointed, jumping quickly from one topic to another without a clear transition. For example, the transition from discussing the nutritional composition of almonds to the production of almond-based beverages could be smoothed out to improve readability.

- Some concepts, such as the stability of almond bagasse powder in relation to humidity, temperature, and water activity, could be explained more clearly to ensure that readers fully understand the study's context.

Materials and Method:

- It would be helpful to provide additional details about the reagents used in the methods, such as brand, purity of reagents, and the exact concentration of prepared solutions.

- The units of measurement for acidity results (g of oleic acid/100 g of dry matter) are correct and clearly specified. However, for the peroxide index, it would be helpful to indicate the unit of measurement for the obtained results (e.g., meq/kg or mEq O2/kg).

Results and discussion

- All tables: Perform all statistical analysis of the results. Typically, the highest value starts with the letter "a"; Standardize decimal places for each parameter; Convert UFC to CFU.

- AFTER REDOING ALL STATISTICAL ANALYSIS, CAREFULLY DISCUSS THE RESULTS.

- Check the y-axis decimal separator and IMPROVE PRESENTATION QUALITY.

- Lack of comparison with other dehydration techniques: The research only compared the effect of hot air drying and freeze-drying on almond bagasse. It would be interesting to include other dehydration techniques, such as infrared drying or microwave drying, to determine which method offers the best preservation of antioxidant compounds and the lowest rate of lipid oxidation. Also, perform an analysis on the industrial application of these two employed techniques.

Author Response

(The authors gave the same response as above.)

Reviewer 3 Report

Comments and Suggestions for Authors

Title - Shelf life of almond bagasse powders as influenced by dehydration and storing conditions

Manuscript Number – foods-2876242

This is a study about almond bagasse drying by different drying methods to see its keeping quality at various storage conditions. It is an interesting work to utilize the almond drink by-product into valuable source of functional components. The authors did hard work to justify the results obtained by drying methods or storage conditions. However, following points are suggested to modify before it is accepted for publication is Foods:

Abstract: The first sentence should be modified to make it clear. For example, Almond bagasse resulting after the production of almond-based drinks represents a promising by-product with potential for use as a functional ingredient.

Introduction: The authors should explain the reason behind to choose hot air drying at 60 and 70 oC along with freeze drying for the drying purpose - The hypothetical reasons to choose these drying techniques and hot air drying temperature.

Line 99 (Evaluation of storage stability of the almond bagasse powder): The authors should provide the detailed specification of aluminum-laminated polyethylene bags used to pack the almond bagasse for storage.

Line 103-106: What is the technical reason to choose such frequency – 15 days for water activity, moisture,….. peroxide value; 1 month for microbiological analysis; and 3 months (90 days) for phenolic compounds?

Line 117-128: L*, a*, b*, ∆E – all of them should be italicized.

Line 189: “an injection volume of 0.5 mL/min and a flow rate of 10 µL at 30 °C.” should be “an injection volume of 10 µL and a flow rate of 0.5 mL/min at 30 °C.”

Line 195: “mg/100 dm” should be “mg/100 g dm”

For all p < 0.05 and p > 0.05, p should be italicized. Also, p ≤ 0.05 is not valid; it should be p < 0.05.

3. Results and discussion section should be divided into different sub-section, based on the methods section.

Line 223: “was reduced to below 0.3, which is the recommended threshold for ensuring” the threshold value may be 0.6? Please confirm this fact from reliable sources.

Line 341-342:No significance difference should be presented as p > 0.05; not p ≤ 0.05.

English writing and grammar should be thoroughly checked by a professional manuscript writer for the minor but blunt mistakes / typos.

Comments on the Quality of English Language

Moderate English correction is necessary from a professional manuscript writer.

Author Response

(The authors gave the same response as above.)

Reviewer 4 Report

Comments and Suggestions for Authors

This paper is dedicated to examining the effect of two types of dehydration (hot air drying and freeze drying) and storage at room temperature and accelerated conditions on the functional quality and stability of almond bagasse powder. the paper is interesting, well structured and explained, but there are some objections:

State in the title that the functional quality of almond bagasse powder was also examined.

In the abstract: Insert numerical results related to the main findings of the work, what detected and highest values.

Line 26, write the latin name in italics.

Line 75, considering that there are many other types of drying, why exactly these two were chosen and compared throughout the paper?

Line 94, Liophilization conditions? Pressure, temperature and duration?

Line 103-106 Why different analyzes were done at different time intervals?

Table 1, 2, 3 water activity and humidity, write the whole word as for the others parameters, and not the marks.

Table 1,2,3 Before the table, write only the title and below the description that goes with the table.

Line 344-346 according to which regulation the safety requirements are met?

In conclusion: emphasize the main finding and innovation that the work brings.

Author Response

(The authors gave the same response as above.)

Round 2

Reviewer 1 Report

Comments and Suggestions for Authors

The manuscript has an interesting topic that has relevance for the industry practice as well. Authors have revised the manuscript thoroughly according to reviewers' comments and suggestions and provided detailed answers for the question of the reviewers. Amendments, rephrasings, more detailed methodology/discussion parts, and improved figure help to achieve a higher scientific quality of the manuscript and make ot more complete. I agree and accept all modifications made by the authors.

Reviewer 2 Report

Comments and Suggestions for Authors

The authors made the suggested corrections. Don't have any more questions.

Reviewer 3 Report

Comments and Suggestions for Authors

Title - Shelf life of almond bagasse powders as influenced by dehydration and storing conditions

Manuscript Number – foods-2876242

The revised version has been assessed and found that the authors have addressed all the issues raised during previous revision and modified satisfactorily; so, this manuscript is recommended for publication in Foods!